# Reinforcement Learning on Web Interfaces using Workflow-Guided Exploration

**Evan Zheran Liu**[†][*]**, Kelvin Guu**[‡][*]**, Panupong Pasupat**[†][*]**, Tianlin Shi**[†]**, Percy Liang**[†]

[†]Department of Computer Science, [‡]Department of Statistics
Stanford University, Stanford, CA 94305, USA
`{evanliu,kguu,ppasupat,tianlins}@stanford.edu,pliang@cs.stanford.edu`

## ABSTRACT

Reinforcement learning (RL) agents improve through trial-and-error, but when reward is sparse and the agent cannot discover successful action sequences, learning stagnates. This has been a notable problem in training deep RL agents to perform web-based tasks, such as booking flights or replying to emails, where a single mistake can ruin the entire sequence of actions. A common remedy is to "warm-start" the agent by pre-training it to mimic expert demonstrations, but this is prone to overfitting. Instead, we propose to *constrain exploration* using demonstrations. From each demonstration, we induce high-level "workflows" which constrain the allowable actions at each time step to be similar to those in the demonstration (e.g., "Step 1: click on a textbox; Step 2: enter some text"). Our exploration policy then learns to identify successful workflows and samples actions that satisfy these workflows. Workflows prune out bad exploration directions and accelerate the agent's ability to discover rewards. We use our approach to train a novel neural policy designed to handle the semi-structured nature of websites, and evaluate on a suite of web tasks, including the recent World of Bits benchmark. We achieve new state-of-the-art results, and show that *workflow-guided exploration* improves sample efficiency over behavioral cloning by more than 100x.

## 1 INTRODUCTION

We are interested in training reinforcement learning (RL) agents to use the Internet (e.g., to book flights or reply to emails) by directly controlling a web browser. Such systems could expand the capabilities of AI personal assistants (Stone & Soper, 2014), which are currently limited to interacting with machine-readable APIs, rather than the much larger world of human-readable web interfaces.

Reinforcement learning agents could learn to accomplish tasks using these human-readable web interfaces through trial-and-error (Sutton & Barto, 1998). But this learning process can be very slow in tasks with sparse reward, where the vast majority of naive action sequences lead to no reward signal (Vecerik et al., 2017; Nair et al., 2017). This is the case for many web tasks, which involve a large action space (the agent can type or click anything) and require a well-coordinated sequence of actions to succeed.

A common countermeasure in RL is to pre-train the agent to mimic expert demonstrations via behavioral cloning (Pomerleau, 1991; Kim et al., 2013), encouraging it to take similar actions in similar states. But in environments with diverse and complex states such as websites, demonstrations may cover only a small slice of the state space, and it is difficult to generalize beyond these states (overfitting). Indeed, previous work has found that warm-starting with behavioral cloning often fails to improve over pure RL (Shi et al., 2017). At the same time, simple strategies to combat overfitting (e.g. using fewer parameters or regularization) cripple the policy's flexibility (Bitzer et al., 2010), which is required for complex spatial and structural reasoning in user interfaces.

In this work, we propose a different method for leveraging demonstrations. Rather than training an agent to directly mimic them, we use demonstrations to *constrain exploration*. By pruning away bad exploration directions, we can accelerate the agent's ability to discover sparse rewards. Furthermore,

---

[*]First three authors contributed equally

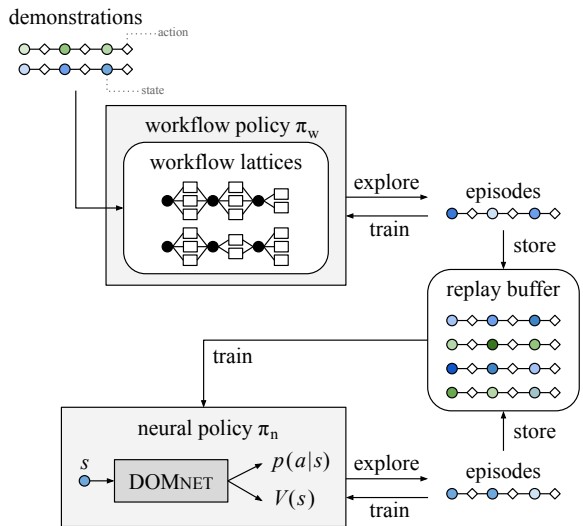

**Preprocessing:**
  **for all** demonstrations $d$ **do**
    Induce workflow lattice from $d$

**Every iteration:**
  Observe an initial environment state
  $\pi_w$ samples a workflow from a lattice
  Roll out an episode $e$ from the workflow
  Use $e$ to update $\pi_w$
  **if** $e$ gets reward $+1$ **then**
    Add $e$ to replay buffer

**Periodically:**
  **if** replay buffer size > threshold **then**
    Sample episodes from replay buffer
    Update $\pi_n$ with sampled episodes
  Observe an initial environment state
  $\pi_n$ rolls out episode $e$
  Update $\pi_n$ and critic $V$ with $e$
  **if** $e$ gets reward $+1$ **then**
    Add $e$ to replay buffer

Figure 1: *Workflow-guided exploration (WGE).* After inducing workflow lattices from demonstrations, the workflow policy $\pi_w$ performs exploration by sampling episodes from sampled workflows. Successful episodes are saved to a replay buffer, which is used to train the neural policy $\pi_n$.

because the agent is not directly exposed to demonstrations, we are free to use a sophisticated neural policy with a reduced risk of overfitting.

To constrain exploration, we employ the notion of a "workflow" (Deka et al., 2016). For instance, given an expert demonstration of how to forward an email, we might infer the following workflow:

$$\text{Click an email title} \rightarrow \text{Click a "Forward" button}$$
$$\rightarrow \text{Type an email address into a textbox} \rightarrow \text{Click a "Send" button}$$

This workflow is more *high-level* than an actual policy: it does not tell us exactly which email to click or which textbox to type into, but it helpfully constrains the set of actions at each time step. Furthermore, unlike a policy, it does not depend on the environment state: it is just a sequence of steps that can be followed blindly. In this sense, a workflow is *environment-blind*. The actual policy certainly should not be environment-blind, but for exploration, we found environment-blindness to be a good inductive bias.

To leverage workflows, we propose the *workflow-guided exploration* (WGE) framework as illustrated in Figure 1:

1. For each demonstration, we extract a lattice of workflows that are consistent with the actions observed in the demonstration (Section 3).

2. We then define a *workflow exploration policy* $\pi_w$ (Section 4), which explores by first selecting a workflow, and then sampling actions that fit the workflow. This policy gradually learns which workflow to select through reinforcement learning.

3. Reward-earning episodes discovered during exploration enter a replay buffer, which we use to train a more powerful and expressive neural network policy $\pi_n$ (Section 5).

A key difference between the web and traditional RL domains such as robotics (Atkeson & Schaal, 1997) or game-playing (Bellemare et al., 2013) is that the state space involves a mix of structured (e.g. HTML) and unstructured inputs (e.g. natural language and images). This motivates us to propose a novel neural network policy (DOMNET), specifically designed to perform flexible relational reasoning over the tree-structured HTML representation of websites.

We evaluate *workflow-guided exploration* and DOMNET on a suite of web interaction tasks, including the MiniWoB benchmark of (Shi et al., 2017), the flight booking interface for Alaska Airlines,

and a new collection of tasks that we constructed to study additional challenges such as noisy environments, variation in natural language, and longer time horizons. Compared to previous results on MiniWoB Shi et al. (2017), which used 10 minutes of demonstrations per task (approximately 200 demonstrations on average), our system achieves much higher success rates and establishes new state-of-the-art results with only 3–10 demonstrations per task.

## 2 SETUP

In the standard reinforcement learning setup, an agent learns a policy $\pi(a|s)$ that maps a state $s$ to a probability distribution over actions $a$. At each time step $t$, the agent observes an environment state $s_t$ and chooses an action $a_t$, which leads to a new state $s_{t+1}$ and a reward $r_t = r(s_t, a_t)$. The goal is to maximize the expected return $\mathbb{E}[R]$, where $R = \sum_t \gamma^t r_{t+1}$ and $\gamma$ is a discount factor. Typical reinforcement learning agents learn through trial-and-error: rolling out episodes $(s_1, a_1, \ldots, s_T, a_T)$ and adjusting their policy based on the results of those episodes.

We focus on settings where the reward is delayed and sparse. Specifically, we assume that (1) the agent receives reward only at the end of the episode, and (2) the reward is high (e.g., $+1$) for only a small fraction of possible trajectories and is uniformly low (e.g., $-1$) otherwise. With large state and action spaces, it is difficult for the exploration policy to find episodes with positive rewards, which prevents the policy from learning effectively.

We further assume that the agent is given a goal $g$, which can either be a structured key-value mapping (e.g., {task: forward, from: Bob, to: Alice}) or a natural language utterance (e.g., *"Forward Bob's message to Alice"*). The agent's state $s$ consists of the goal $g$ and the current state of the web page, represented as a tree of elements (henceforth *DOM tree*). We restrict the action space to click actions `Click(e)` and type actions `Type(e,t)`, where `e` is a leaf element of the DOM tree, and `t` is a string from the goal $g$ (a value from a structured goal, or consecutive tokens from a natural language goal). Figure 2 shows an example episode for an email processing task. The agent receives $+1$ reward if the task is completed correctly, and $-1$ reward otherwise.

## 3 INDUCING WORKFLOWS FROM DEMONSTRATIONS

Given a collection of expert demonstrations $d = (\tilde{s}_1, \tilde{a}_1, \ldots, \tilde{s}_T, \tilde{a}_T)$, we would like explore actions $a_t$ that are "similar" to the demonstrated actions $\tilde{a}_t$. Workflows capture this notion of similarity by specifying a set of similar actions at each time step. Formally, a workflow $z_{1:T}$ is a sequence of workflow steps, where each step $z_t$ is a function that takes a state $s_t$ and returns a constrained set $z_t(s_t)$ of similar actions. We use a simple compositional constraint language (Appendix A) to describe workflow steps. For example, with $z_t = $ `Click(Tag("img"))`, the set $z_t(s_t)$ contains click actions on any DOM element in $s_t$ with tag `img`.

We induce a set of workflows from each demonstration $d = (\tilde{s}_1, \tilde{a}_1, \ldots, \tilde{s}_T, \tilde{a}_T)$ as follows. For each time step $t$, we enumerate a set $Z_t$ of all possible workflow steps $z_t$ such that $\tilde{a}_t \in z_t(\tilde{s}_t)$. The set of workflows is then the cross product $Z_1 \times \cdots \times Z_T$ of the steps. We can represent the induced workflows as paths in a *workflow lattice* as illustrated in Figure 2.

To handle noisy demonstrations where some actions are unnecessary (e.g., when the demonstrator accidentally clicks on the background), we add shortcut steps that skip certain time steps. We also add shortcut steps for any consecutive actions that can be collapsed into a single equivalent action (e.g., collapsing two type actions on the same DOM element into a single `Type` step). These shortcuts allow the lengths of the induced workflows to differ from the length of the demonstration. We henceforth ignore these shortcut steps to simplify the notation.

The induced workflow steps are not equally effective. For example in Figure 2, the workflow step `Click(Near(Text("Bob")))` (Click an element near text "Bob") is too specific to the demonstration scenario, while `Click(Tag("div"))` (Click on any `<div>` element) is too general and covers too many irrelevant actions. The next section describes how the workflow policy $\pi_w$ learns which workflow steps to use.

**Demonstration:** goal = {**task:** forward, **from:** Bob, **to:** Alice}

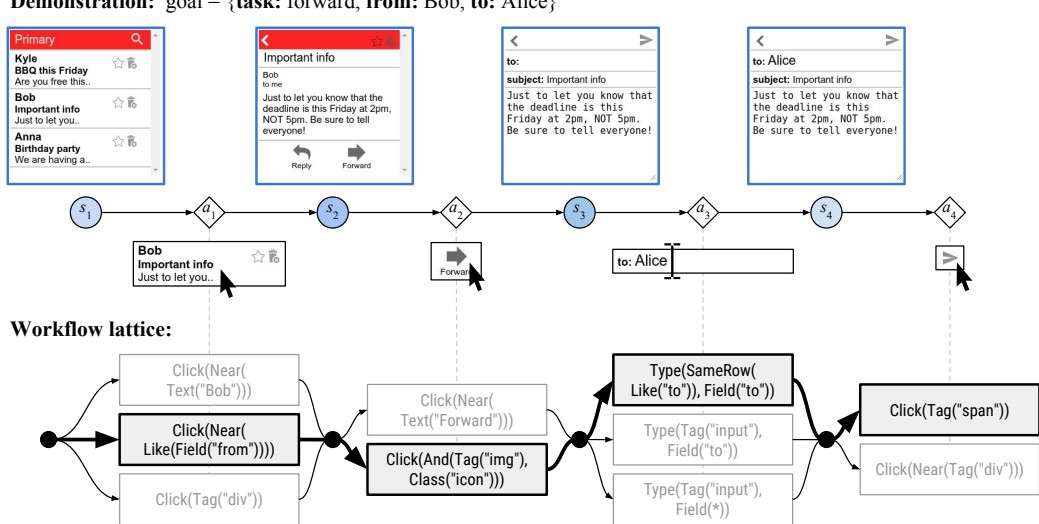

Figure 2: From each demonstration, we induce a workflow lattice based on the actions in that demonstration. Given a new environment, the workflow policy samples a workflow (a path in the lattice, as shown in bold) and then samples actions that fit the steps of the workflow.

## 4  WORKFLOW EXPLORATION POLICY

Our workflow policy interacts with the environment to generate an episode in the following manner. At the beginning of the episode, the policy conditions on the provided goal $g$, and selects a demonstration $d$ that carried out a similar goal:

$$d \sim p(d|g) \propto \exp[\text{sim}(g, g_d)] \tag{1}$$

where $\text{sim}(g, g_d)$ measures the similarity between $g$ and the goal $g_d$ of demonstration $d$. In our tasks, we simply let $\text{sim}(g, g_d)$ be 1 if the structured goals share the same keys, and $-\infty$ otherwise.

Then, at each time step $t$ with environment state $s_t$, we sample a workflow step $z_t$ according to the following distribution:

$$z_t \sim \pi_{\text{w}}(z|d, t) \propto \exp(\psi_{z,t,d}), \tag{2}$$

where each $\psi_{z,t,d}$ is a separate scalar parameter to be learned. Finally, we sample an action $a_t$ uniformly from the set $z_t(s_t)$.

$$a_t \sim p(a|z_t, s_t) = \frac{1}{|z_t(s_t)|} \tag{3}$$

The overall probability of exploring an episode $e = (s_1, a_1, \ldots, s_T, a_T)$ is then:

$$p(e|g) = p(d|g) \prod_{t=1}^{T} p(s_t|s_{t-1}, a_{t-1}) \sum_z p(a_t|z, s_t) \pi_{\text{w}}(z|d, t) \tag{4}$$

where $p(s_t|s_{t-1}, a_{t-1})$ is the (unknown) state transition probability.

Note that $\pi_{\text{w}}(z|d, t)$ is not a function of the environment states $s_t$ at all. Its decisions only depend on the selected demonstration and the current time $t$. This *environment-blindness* means that the workflow policy uses far fewer parameters than a state-dependent policy, enabling it to learn more quickly and preventing overfitting. Due to *environment-blindness*, the workflow policy cannot solve the task, but it quickly learns to certain good behaviors, which can help the neural policy learn.

To train the workflow policy, we use a variant of the REINFORCE algorithm (Williams, 1992; Sutton & Barto, 1998). In particular, after rolling out an episode $e = (s_1, a_1, \ldots, s_T, a_T)$, we approximate the gradient using the unbiased estimate

$$\sum_t (G_t - v_{d,t}) \nabla_\psi \log \sum_z p(a_t|z, s_t) \pi_{\text{w}}(z|d, t), \tag{5}$$

where $G_t$ is the return at time step $t$ and $v_{d,t}$ is a baseline term for variance reduction.

Sampled episodes from the workflow policy that receive a positive reward are stored in a replay buffer, which will be used for training the neural policy $\pi_n$.

# 5 NEURAL POLICY

As outlined in Figure 1, the neural policy is learned using both on-policy and off-policy updates (where episodes are drawn from the replay buffer). Both updates use A2C, the synchronous version of the advantage actor-critic algorithm (Mnih et al., 2016). Since only episodes with reward +1 enter the replay buffer, the off-policy updates behave similarly to supervised learning on optimal trajectories. Furthermore, successful episodes discovered during on-policy exploration are also added to the replay buffer.

**Model architecture.** We propose DOMNET, a neural architecture that captures the spatial and hierarchical structure of the DOM tree. As illustrated in Figure 5, the model first embeds the DOM elements and the input goal, and then applies a series of attentions on the embeddings to finally produce a distribution over actions $\pi_n(a|s)$ and a value function $V(s)$, the critic. We highlight our novel DOM embedder, and defer other details to Appendix C.

We design our DOM embedder to capture the various interactions between DOM elements, similar to recent work in graph embeddings (Kipf & Welling, 2017; Pham et al., 2017; Hamilton et al., 2017). In particular, DOM elements that are "related" (e.g., a checkbox and its associated label) should pass their information to each other.

To embed a DOM element $e$, we first compute the *base embedding* $v_{\text{base}}^e$ by embedding and concatenating its attributes (tag, classes, text, etc.). In order to capture the relationships between DOM elements, we next compute two types of *neighbor embeddings*:

1. We define *spatial neighbors* of $e$ to be any element $e'$ within 30 pixels from $e$, and then sum up their base embeddings to get the *spatial neighbor embedding* $v_{\text{spatial}}^e$.

2. We define *depth-$k$ tree neighbors* of $e$ to be any element $e'$ such that the least common ancestor of $e$ and $e'$ in the DOM tree has depth at most $k$. Intuitively, tree neighbors of a higher depth are more related. For each depth $k$, we apply a learnable affine transformation $f$ on the base embedding of each depth-$k$ tree neighbor $e'$, and then apply max pooling to get $v_{\text{tree}[k]}^e = \max f(v_{\text{base}}^{e'})$. We let the *tree neighbor embedding* $v_{\text{tree}}^e$ be the concatenation of $v_{\text{tree}[k]}^e$ for $k = 3, 4, 5, 6$.

Finally, we define the *goal matching embedding* $v_{\text{match}}^e$ to be the sum of the embeddings of all words in $e$ that also appear in the goal. The final embedding $v_{\text{DOM}}^e$ of $e$ is the concatenation of the four embeddings $[v_{\text{base}}^e; v_{\text{spatial}}^e; v_{\text{tree}}^e; v_{\text{match}}^e]$.

# 6 EXPERIMENTS

## 6.1 TASK SETUPS

We evaluate our approach on three suites of interactive web tasks:

1. *MiniWoB*: the MiniWoB benchmark of Shi et al. (2017)

2. *MiniWoB++*: a new set of tasks that we constructed to incorporate additional challenges not present in MiniWoB, such as stochastic environments and variation in natural language.

3. *Alaska*: the mobile flight booking interface for Alaska Airlines, inspired by the FormWoB benchmark of Shi et al. (2017).

We describe the common task settings of the MiniWoB and MiniWoB++ benchmarks, and defer the description of the Alaska benchmark to Section 6.3.3.

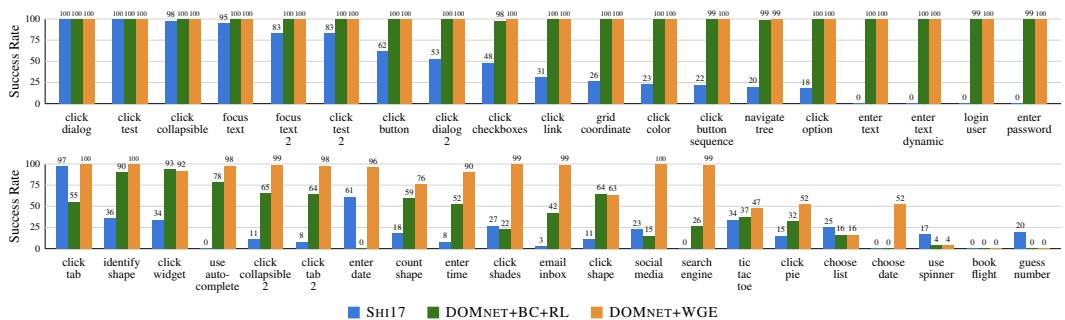

Figure 3: Success rates of different approaches on the MiniWoB tasks. DOMNET+WGE outperforms SHI17 on all but two tasks and effectively solves a vast majority.

| Task | Description | Steps | BC+RL | $\pi_w$ only | WGE |
|---|---|---|---|---|---|
| click-checkboxes | Click 0–6 specified checkboxes | 7 | 98 | 81 | **100** |
| click-checkboxes-large[+] | . . . 5–12 targets | 13 | 0 | 43 | **84** |
| click-checkboxes-soft[+] | . . . specifies synonyms of the targets | 7 | 51 | 34 | **94** |
| click-checkboxes-transfer[+] | . . . training data has 0-3 targets | 7 | **64** | 17 | **64** |
| multi-ordering[+] | Fill a form with varying field orderings | 4 | 5 | 78 | **100** |
| multi-layout[+] | Fill a form with varying UIs layouts | 4 | 99 | 9 | **100** |
| social-media | Do an action on the specified Tweet | 2 | 15 | 2 | **100** |
| social-media-all[+] | . . . on all matching Tweets | 12 | **1** | 0 | 0 |
| social-media-some[+] | . . . on specified no. of matching Tweets | 12 | 2 | 3 | **42** |
| email-inbox | Perform tasks on an email inbox | 4 | 43 | 3 | **99** |
| email-inbox-nl[+] | . . . natural language goal | 4 | 28 | 0 | **93** |

Table 1: Results on additional tasks. (+ = MiniWoB++, Steps = task length as the maximum number of steps needed for a perfect policy to complete the task)

**Environment.** Each task contains a 160px × 210px environment and a goal specified in text. The majority of the tasks return a single sparse reward at the end of the episode; either +1 (success) or −1 (failure). For greater consistency among tasks, we disabled *all* partial rewards in our experiments. The agent has access to the environment via a Selenium web driver interface.

The public MiniWoB benchmark[1] contains 80 tasks. We filtered for the 40 tasks that only require actions in our action space, namely clicking on DOM elements and typing strings from the input goal. Many of the excluded tasks involve somewhat specialized reasoning, such as being able to compute the angle between two lines, or solve algebra problems. For each task, we used Amazon Mechanical Turk to collect 10 demonstrations, which record all mouse and keyboard events along with the state of the DOM when each event occurred.

**Evaluation metric.** We report *success rate*: the percentage of test episodes with reward +1. Since we have removed partial rewards, success rate is a linear scaling of the average reward, and is equivalent to the definition of success rate in Shi et al. (2017).

## 6.2 MAIN RESULTS

We compare the success rates across the MiniWoB tasks of the following approaches:

- SHI17: the system from Shi et al. (2017), pre-trained with behavioral cloning on 10 minutes of demonstrations (approximately 200 demonstrations on average) and fine-tuned with RL. Unlike DOMNET, this system primarily uses a pixel-based representation of the state.[2]

---

[1] http://alpha.openai.com/miniwob/

[2] It is augmented with filters that activate on textual elements which overlap with goal text.

- DOMNET+BC+RL: our proposed neural policy, DOMNET, but pre-trained with behavioral cloning on 10 demonstrations and fine-tuned with RL, like SHI17. During behavioral cloning, we apply early stopping based on the reward on a validation set.
- DOMNET+WGE: our proposed neural policy, DOMNET, trained with workflow-guided exploration on 10 demonstrations.

For DOMNET+BC+RL and DOMNET+WGE, we report the test success rate at the time step where the success rate on a validation set reaches its maximum.

The results are shown in Figure 3. By comparing SHI17 with DOMNET+BC+RL, we can roughly evaluate the contribution of our new neural architecture DOMNET, since the two share the same training procedure (BC+RL). While SHI17 also uses the DOM tree to compute text alignment features in addition to the pixel-level input, our DOMNET uses the DOM structure more explicitly. We find DOMNET+BC+RL to empirically improve the success rate over SHI17 on most tasks.

By comparing DOMNET+BC+RL and DOMNET+WGE, we find that workflow-guided exploration enables DOMNET to perform even better on the more difficult tasks, which we analyze in the next section. Some of the workflows that the workflow policy $\pi_w$ learns are shown in Appendix B.

## 6.3 ANALYSIS

### 6.3.1 MINIWOB++ BENCHMARK

We constructed and released the MiniWoB++ benchmark of tasks to study additional challenges a web agent might encounter, including: longer time horizons (click-checkboxes-large), "soft" reasoning about natural language (click-checkboxes-soft), and stochastically varying layouts (multi-orderings, multi-layouts). Table 1 lists the tasks and their time horizons (number of steps needed for a perfect policy to carry out the longest goal) as a crude measure of task complexity.

We first compare the performance of DOMNET trained with BC+RL (baseline) and DOMNET trained with WGE (our full approach). The proposed WGE model outperforms the BC+RL model by an average of 42% absolute success rate. We analyzed their behaviors and noticed two common failure modes of training with BC+RL that are mitigated by instead training with WGE:

1. The BC+RL model has a tendency to take actions that prematurely terminate the episode (e.g., hitting "Submit" in click-checkboxes-large before all required boxes are checked). One likely cause is that these actions occur across all demonstrations, while other non-terminating actions (e.g., clicking different checkboxes) vary across demonstrations.
2. The BC+RL model occasionally gets stuck in cyclic behavior such as repeatedly checking and unchecking the same checkbox. These failure modes stem from overfitting to parts of the demonstrations, which WGE avoids.

Next, we analyze the workflow policy $\pi_w$ learned by WGE. The workflow policy $\pi_w$ by itself is too simplistic to work well at test time for several reasons:

1. Workflows ignore environment state and therefore cannot respond to the differences in the environment, such as the different layouts in multi-layouts.
2. The workflow constraint language lacks the expressivity to specify certain actions, such as clicking on synonyms of a particular word in click-checkboxes-soft.
3. The workflow policy lacks expressivity to select the correct workflow for a given goal.

Nonetheless the workflow policy $\pi_w$ is sufficiently constrained to discover reward some of the time, and the neural policy $\pi_n$ is able to learn the right behavior from such episodes. As such, the neural policy can achieve high success rates even when the workflow policy $\pi_w$ performs poorly.

### 6.3.2 NATURAL LANGUAGE INPUTS

While MiniWoB tasks provide structured goals, we can also apply our approach to natural language goals. We collected a training dataset using the overnight data collection technique (Wang et al., 2015). In the email-inbox-nl task, we collected natural language templates by asking annotators

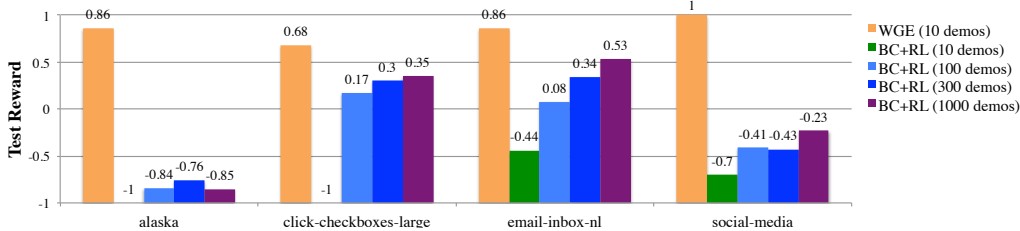

Figure 4: Comparison between DOMNET+BC+RL and DOMNET+WGE on several of the most difficult tasks, evaluated on test reward. DOMNET+WGE trained on 10 demonstrations outperforms DOMNET+BC+RL even with 1000 demonstrations.

to paraphrase the task goals (e.g., *"Forward Bob's message to Alice"* → *"Email Alice the email I got from Bob"*) and then abstracting out the fields (*"Email <TO> the email I got from <FROM>"*). During training, the workflow policy $\pi_w$ receives states with both the structured goal and the natural language utterance generated from a random template, while the neural policy $\pi_n$ receives only the utterance. At test time, the neural policy is evaluated on unseen utterances. The results in Table 1 show that the WGE model can learn to understand natural language goals (93% success rate).

Note that the workflow policy needs access to the structured inputs only because our constraint language for workflow steps operates on structured inputs. The constraint language could potentially be modified to work with utterances directly (e.g., `After("to")` extracts the utterance word after *"to"*), but we leave this for future work.

### 6.3.3 SCALING TO REAL WORLD TASKS

We applied our approach on the Alaska benchmark, a more realistic flight search task on the Alaska Airlines mobile site inspired by the FormWoB task in Shi et al. (2017). In this task, the agent must complete the flight search form with the provided information (6–7 fields). We ported the web page to the MiniWoB framework with a larger 375px × 667px screen, replaced the server backend with a surrogate JavaScript function, and clamped the environment date to March 1, 2017.

Following Shi et al. (2017), we give partial reward based on the fraction of correct fields in the submitted form if all required fields are filled in. Despite this partial reward, the reward is still extremely sparse: there are over 200 DOM elements (compared to ≈ 10–50 in MiniWoB tasks), and a typical episode requires at least 11 actions involving various types of widgets such as autocompletes and date pickers. The probability that a random agent gets positive reward is less than $10^{-20}$.

We first performed experiments on Alaska-Shi17, a clone of the original Alaska Airlines task in Shi et al. (2017), where the goal always specifies a roundtrip flight (two airports and two dates). On their dataset, our approach, using only 1 demonstration, achieves an average reward of 0.97, compared to their best result of 0.57, which uses around 80 demonstrations.

Our success motivated us to test on a more difficult version of the task which additionally requires selecting flight type (a checkbox for one-way flight), number of passengers (an increment-decrement counter), and seat type (hidden under an accordion). We achieve an average reward of 0.86 using 10 demonstrations. This demonstrates our method can handle long horizons on real-world websites.

### 6.3.4 SAMPLE EFFICIENCY

To evaluate the demonstration efficiency of our approach, we compare DOMNET+WGE with DOMNET+BC+RL trained on increased numbers of demonstrations. We compare DOMNET+WGE trained on 10 demonstrations with DOMNET+BC+RL on 10, 100, 300, and 1000 demonstrations. The test rewards[3] on several of the hardest tasks are summarized in Figure 4.

Increasing the number of demonstrations improves the performance of BC+RL, as it helps prevent overfitting. However, on every evaluated task, WGE trained with only 10 demonstrations still achieves much higher test reward than BC+RL with 1000 demonstrations. This corresponds to an

---

[3]We report test reward since success rate is artificially high in the Alaska task due to partial rewards.

over 100x sample efficiency improvement of our method over behavioral cloning in terms of the number of demonstrations.

# 7 DISCUSSION

**Learning agents for the web.**    Previous work on learning agents for web interactions falls into two main categories. First, simple programs may be specified by the user (Yeh et al., 2009) or may be inferred from demonstrations (Allen et al., 2007). Second, soft policies may be learned from scratch or "warm-started" from demonstrations (Shi et al., 2017). Notably, sparse rewards prevented Shi et al. (2017) from successfully learning, even when using a moderate number of demonstrations. While policies have proven to be more difficult to learn, they have the potential to be expressive and flexible. Our work takes a step in this direction.

**Sparse rewards without prior knowledge.**    Numerous works attempt to address sparse rewards without incorporating any additional prior knowledge. Exploration methods (Osband et al., 2016; Chentanez et al., 2005; Weber et al., 2017) help the agent better explore the state space to encounter more reward; shaping rewards (Ng et al., 1999) directly modify the reward function to encourage certain behaviors; and other works (Jaderberg et al., 2016; Andrychowicz et al., 2017) augment the reward signal with additional unsupervised reward. However, without prior knowledge, helping the agent receive additional reward is difficult in general.

**Imitation learning.**    Various methods have been proposed to leverage additional signals from experts. For instance, when an expert policy is available, methods such as DAGGER (Ross et al., 2011) and AGGREVATE (Ross & Bagnell, 2014; Sun et al., 2017) can query the expert policy to augment the dataset for training the agent. When only expert *demonstrations* are available, inverse reinforcement learning methods (Abbeel & Ng, 2004; Ziebart et al., 2008; Finn et al., 2016; Ho & Ermon, 2016; Baram et al., 2017) infer a reward function from the demonstrations without using reinforcement signals from the environment.

The usual method for incorporating both demonstrations and reinforcement signals is to pre-train the agent with demonstrations before applying RL. Recent work extends this technique by (1) introducing different objective functions and regularization during pre-training, and (2) mixing demonstrations and rolled-out episodes during RL updates (Hosu & Rebedea, 2016; Hester et al., 2018; Vecerik et al., 2017; Nair et al., 2017).

Instead of training the agent on demonstrations directly, our work uses demonstrations to *guide exploration*. The core idea is to explore trajectories that lie in a "neighborhood" surrounding an expert demonstration. In our case, the neighborhood is defined by a workflow, which only permits action sequences analogous to the demonstrated actions. Several previous works also explore neighborhoods of demonstrations via reward shaping (Brys et al., 2015; Hussein et al., 2017) or off-policy sampling (Levine & Koltun, 2013). One key distinction of our work is that we define neighborhoods in terms of action similarity rather than state similarity. This distinction is particularly important for the web tasks: we can easily and intuitively describe how two actions are analogous (e.g., "they both type a username into a textbox"), while it is harder to decide if two web page states are analogous (e.g., the email inboxes of two different users will have completely different emails, but they could still be analogous, depending on the task.)

**Hierarchical reinforcement learning.**    Hierarchical reinforcement learning (HRL) methods decompose complex tasks into simpler subtasks that are easier to learn. Main HRL frameworks include abstract actions (Sutton et al., 1999; Konidaris & Barto, 2007; Hauser et al., 2008), abstract partial policies (Parr & Russell, 1998), and abstract states (Roderick et al., 2017; Dietterich, 1998; Li et al., 2006). These frameworks require varying amounts of prior knowledge. The original formulations required programmers to manually specify the decomposition of the complex task, while Andreas et al. (2016) only requires supervision to identify subtasks, and Bacon et al. (2017); Daniel et al. (2016) learn the decomposition fully automatically, at the cost of performance.

Within the HRL methods, our work is closest to Parr & Russell (1998) and the line of work on constraints in robotics (Phillips et al., 2016; Perez-D'Arpino & Shah, 2017). The work in Parr & Russell (1998) specifies partial policies, which constrain the set of possible actions at each state,

similar to our workflow items. In contrast to previous instantiations of the HAM framework (Andre, 2003; Marthi & Guestrin, 2005), which require programmers to specify these constraints manually, our work automatically induces constraints from user demonstrations, which do not require special skills to provide. Phillips et al. (2016); Perez-D'Arpino & Shah (2017) also resemble our work, in learning constraints from demonstrations, but differ in the way they use the demonstrations. Whereas our work uses the learned constraints for exploration, Phillips et al. (2016) only uses the constraints for planning and Perez-D'Arpino & Shah (2017) build a knowledge base of constraints to use at test time.

**Summary.** Our workflow-guided framework represents a judicious combination of demonstrations, abstractions, and expressive neural policies. We leverage the targeted information of demonstrations and the inductive bias of workflows. But this is only used for exploration, protecting the expressive neural policy from overfitting. As a result, we are able to learn rather complex policies from a very sparse reward signal and very few demonstrations.

**Acknowledgments.** This work was supported by NSF CAREER Award IIS-1552635.

**Reproducibility.** Our code and data are available at `https://github.com/stanfordnlp/wge`. Reproducible experiments are available on the CodaLab platform at `https://worksheets.codalab.org/worksheets/0x0f25031bd42f4aabbc17625fe1484066/`.

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

## A  CONSTRAINT LANGUAGE FOR WORKFLOW STEPS

We try to keep the constraint language as minimal and general as possible. The main part of the language is the object selector (*elementSet*) which selects either (1) objects that share a specified property, or (2) objects that align spatially. These two types of constraints should be applicable in many typical RL domains such as game playing and robot navigation.

| | | |
|---|---|---|
| *constraint* | ::= | `Click(`*elementSet*`)` |
| | | [Any click action on an element in *elementSet*] |
| | \| | `Type(`*elementSet,string*`)` |
| | | [Any type action that types *string* on an element in *elementSet*] |
| | \| | `Type(`*elementSet,*`Field(*))` |
| | | [Any type action that types a goal field value on an element in *elementSet*] |
| *elementSet* | ::= | `Tag(`*tag*`)` |
| | | [Any element with HTML tag *tag*] |
| | \| | `Text(`*string*`)` |
| | | [Any element with text *string*] |
| | \| | `Like(`*string*`)` |
| | | [Any element whose text is a substring of *string*] |
| | \| | `Near(`*elementSet*`)` |
| | | [Any element that is within 30px from an element in *elementSet*] |
| | \| | `SameRow(`*elementSet*`)` |
| | | [Any element that aligns horizontally with an element in *elementSet*] |
| | \| | `SameCol(`*elementSet*`)` |
| | | [Any element that aligns vertically with an element in *elementSet*] |
| | \| | `And(`*elementSet,*`Class(*classes*))` |
| | | [Any element from *elementSet* matching some class name in *classes*] |
| *tag* | ::= | a valid HTML tag name |
| *string* | ::= | a string literal |
| | \| | `Field(`*fieldName*`)` |
| | | [The value from the goal field *fieldName*] |
| *classes* | ::= | a list of valid HTML class names |

To avoid combinatorial explosion of relatively useless constraints, we limit the number of nested *elementSet* applications to 3, where the third application must be the `Class` filter. When we induce workflow steps from a demonstration, the valid literal values for *tag*, *string*, and *classes* are extracted from the demonstration state.

## B  EXAMPLES OF LEARNED WORKFLOWS

**login-user**
*Enter the username "ashlea" and password "k0UQp" and press login.*
{username: ashlea, password: k0UQp}

```
Type(Tag("input_text"),Field("username"))
Type(Tag("input_password"),Field("password"))
            Click(Like("Login"))
```

**email-inbox**
*Find the email by Ilka and forward it to Krista.*
{task: forward, name: Ilka, to: Krista}

```
            Click(Near(Field("by")))
            Click(SameCol(Like("Forward")))
Type(And(Near("Subject"),Class("forward-sender")),Field("to"))
            Click(Tag("span"))
```

**search-engine**

*Enter "Cheree" and press "Search", then find and click the 5th search result.*

{target: Cheree, rank: 5}

```
Type(Near(Tag("button")),Field(*))
        Click(Text("Search"))
          Click(Like(">"))
      Click(Text(Field("target")))
```

**Alaska**

{departure city: Tampa, destination city: Seattle, ticket type: return flight,
departure day: 6, returning Day: 16, passengers: 3, seat type: first }

```
Type(And(Near(Like("From")),Class("text-input-pad")),Field("departure city"))
    Click(And(SameRow(Tag("label")),Class(["input-selection","last"])))
Type(And(Near(Like("To")),Class("text-input-pad")),Field("destination city"))
              Click(Like(Field("destination city")))
      Click(And(SameCol(Tag("a")),Class(["calbg","text-input"])))
            Click(Text(Field("departure day")))
                    Click(Like("Done"))
      Click(Near(Like("Return")),Class(["calbg","text-input"]))
            Click(Text(Field("returning day")))
                    Click(Like("Done"))
                     Click(Like("+"))
                     Click(Like("+"))
                     Click(Tag("h2"))
                    Click(Text("First"))
          Click(And(Near(Tag("body")),Class("button")))
```

## C  DETAILS OF THE NEURAL MODEL ARCHITECTURE

**Embeddings.**  From the input state, we first embed the DOM elements $e$ and the goal units $u$, where $u$ is a key-value pair for structured goals and a token for natural language goals.

The process for computing the embedding $v^e_{\text{DOM}}$ of DOM elements is already described in Section 5. For the goal unit embedding $v^u_{\text{goal}}$, we embed each key-value pair as the sum of word embeddings, and embed natural language goals with an LSTM.

**Attentions.**  After obtaining the embedding $v^e_{\text{DOM}}$ of each DOM element $e$ and $v^u_{\text{goal}}$ of each goal unit $u$, we apply a series of attentions to relate the DOM elements with the goal:

1. *DOM context*: we applied max-pooling on $v^e_{\text{DOM}}$ to get a query vector, and then attend over the DOM embeddings $v^e_{\text{DOM}}$. The DOM context is the weighted average of the attended DOM embeddings.

2. *Goal contexts*: we use the DOM context as the query vector to attend over the goal embeddings $v^u_{\text{goal}}$. We compute two goal contexts from two different attention heads. Each head uses sentinel attention, where part of the attention can be put on a learned NULL vector, which is useful for ignoring the goal when the next action should not depend on the goal.

3. *DOM element selection*: We concatenate the DOM context and goal contexts into a query vector to attend over on the DOM embeddings $v^e_{\text{DOM}}$. We use two attention heads, and combine the attention weights from the two heads based on ratio computed from the goal contexts. The result is a distribution over the target DOM elements e.

4. *Typed string and action selection*: For a given target DOM element e, we combine the goal context and the embedding $v^e_{\text{DOM}}$ of e to get a query vector to attend over the goal embeddings $v^u_{\text{goal}}$. For structured queries, we get a distribution over the goal fields, while for natural language queries, we get distributions of the start and end tokens. The same query vector is also used to compute the distribution over the action types (click or type).

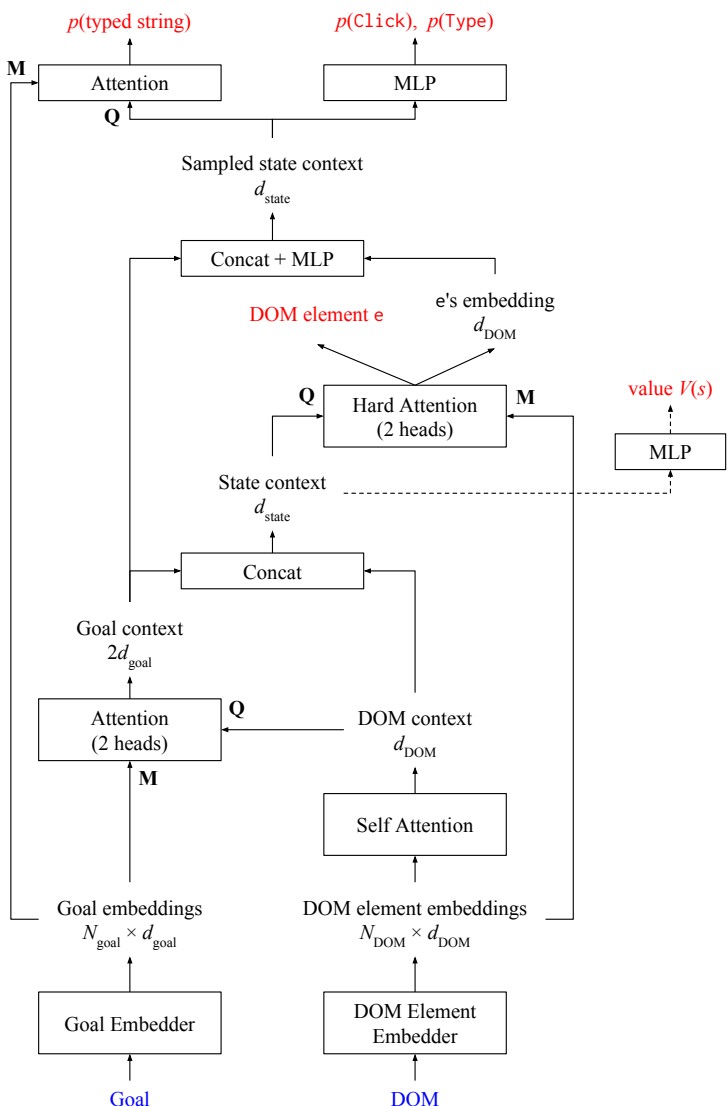

Figure 5: The architecture of the neural policy $\pi_n$. The inputs from the state are denoted in blue, while the outputs are denoted in red. **Q** = query vector; **M** = memory matrix.

