# OpenReview forum: "Reinforcement Learning on Web Interfaces using Workflow-Guided Exploration"
_ICLR.cc/2018/Conference — Accept (Poster)_

### Official Review · AnonReviewer2 · 2017-11-17
**The paper has good empirical results, but algorithmic novelty is small.  It could benefit from more concrete comparisons in the literaure**

**Rating:** 6
**Confidence:** 3

**Review:**

SUMMARY

The paper deals with the problem of training RL algorithms from demonstration and applying them to various web interfaces such as booking flights.  Specifically, it is applied to the Mini world of Bids benchmark (http://alpha.openai.com/miniwob/).

The difference from existing work is that rather than training an agent to directly mimic the demonstrations, it uses demonstrations to constrain exploration. By pruning away bad exploration directions.

The idea is to  build a lattice of workflows from demonstration and randomly sample sequence of actions from this lattice that satisfy the current goal.    Use the sequences of actions to sample trajectories and use the trajectories to learn the RL policy.



COMMENTS


In effect, the workflow sequences provide more generalization than simply mimicking, but It not obvious, why they don’t run into overfitting problems.  However experimentally the paper performs better than the previous approach.

There is a big literature on learning from demonstrations that the authors could compare with, or explain why their work is different.

In addition, they make general comparison to RL literature such as hierarchy rather than more concrete comparisons with the problem at hand (learning from demonstrations.)

What does DOM stand for?  The paper is not self-contained.  For example, what does DOM stand for?


In the results of table 1 and Figure 3.  Why more steps mean success?

In equation 4 there seems to exist an environment model.  Why do we need to use this whole approach in the paper then?  Couldn’t  we just do policy iteration?

---

> ### Author Response · Authors · 2017-12-04
> **Reply to AnonReviewer2**
>
> We would like to thank the reviewer for their detailed and thoughtful feedback.
>
> In our revision, we will significantly expand our comparison to related work on learning from demonstrations. The summary is provided below:
>
> Previous work on learning from demonstration fall into two broad categories:
>     1) Using demonstrations to directly update policy parameters (e.g., behavioral cloning, IRL, etc.).
>     2) Using demonstrations to guide or constrain exploration.
>
> Our method belongs to category (2). The core idea is to explore trajectories that lie in a "neighborhood" surrounding an expert demonstration. In our work, the neighborhood is defined by a workflow, which only permits action sequences analogous to the demonstrated actions.
>
> Other methods in category (2) also explore the neighborhood surrounding a demonstration, using shaping rewards (Brys et al. 2015, Hussein et al. 2017) or off-policy sampling (Levine & Koltun, 2013). A key difference is that we define our neighborhood in terms of action-similarity, rather than state-similarity. This distinction is particularly important for the web tasks we study: we can easily and intuitively describe how two actions are analogous (e.g., "they both type a username into a textbox"), while it is harder to decide if two web page states are analogous (e.g., the email inboxes of two different users will have completely different emails, but they could still be analogous, depending on the task.)
>
> Regarding overfitting: our workflow policy does not overfit to demonstrations because the demonstrations are merely used to induce a workflow lattice -- the actual parameters of the workflow policy are learned through trial-and-error reinforcement learning, for which there is infinite data.
>
> The workflow policy maintains a distribution over possible workflows. Some workflows define a very small neighborhood of trajectories surrounding an expert demonstration (tight), while others impose almost no constraints (loose). As the workflow policy is trained, it converges to the tightest workflow that can successfully generalize.
>
> Our final neural policy also does not overfit, because it is trained on a replay buffer of successful episodes discovered by the workflow policy, which is much larger than the original set of demonstrations.
>
> We would like to also quickly address the other questions, which we will be sure to clarify in the paper:
>     - In Equation 4, we do not have access to the environment model p(s_t | s_{t-1}, a_{t-1}). We merely state that our
>       sampling procedure produces episodes e following the distribution p(e | g) of Equation 4, which we do not
>       compute.
>     - DOM stands for Document Object Model, the standard tree-based representation of a web page.
>     - The "Steps" column in Table 1 is the number of steps needed to complete the task under the optimal behavior
>       (e.g., by a human expert). It is a rough measure of task difficulty and is not related to model performance.

---

### Official Review · AnonReviewer3 · 2017-11-27
**Interesting, but hard to know the novelty**

**Rating:** 7
**Confidence:** 3

**Review:**

This paper introduces a new exploration policy for Reinforcement Learning for agents on the web called "Workflow Guided Exploration". Workflows are defined through a DSL unique to the domain.

The paper is clear, very well written, and well-motivated. Exploration is still a challenging problem for RL. The workflows remind me of options though in this paper they appear to be hand-crafted. In that sense, I wonder if this has been done before in another domain. The results suggest that WGE sometimes helps but not consistently. While the experiments show that DOMNET improves over Shi et al, that could be explained as not having to train on raw pixels or not enough episodes.

---

> ### Author Response · Authors · 2017-12-04
> **Reply to AnonReviewer3**
>
> We would like to thank the reviewer for their helpful feedback!
>
> The key distinction between our work and most existing hierarchical RL approaches (e.g., options, MAXQ) is that our hierarchical structures (workflows) are inferred from demonstrations, rather than manually crafted or learned from scratch.
>
> We try to keep the constraint language for describing workflow steps as minimal and general as possible. The main part of the language is just an element selector (elementSet) which selects either (1) things that share a specified property, or (2) things that align spatially, both of which are applicable in many typical RL domains (game playing, robot navigation, etc.)
>
> In our experiments (Figure 3), WGE (red) consistently performs equally or better than behavioral cloning (green). There are some easy tasks in the benchmark (e.g., click-button), where both WGE and the baselines have perfect performance. But in more difficult tasks (Table 1), WGE greatly improves over baselines by an average of 42% absolute success rate.
>
> Regarding the comparison with Shi17:
>         - Shi17 used ~200 demonstrations per task, whereas we achieve superior performance with only 3-10.
>         - In addition to pixel-level data, the model of Shi17 actually also uses the DOM tree to compute text alignment
>           features. Our DOMNet uses the DOM structure more explicitly, which indeed produces better performance.
>         - Our DOMNet+BC+RL baseline separates the contribution of DOMNet from Workflow-Guided Exploration. Table 1
>           and Figure 3 illustrate that both are important.

---

### Official Review · AnonReviewer1 · 2017-12-15
**WGE seems cool but why not have an IL or IRL baseline?**

**Rating:** 7
**Confidence:** 4

**Review:**

Summary:

The authors propose a method to make exploration in really sparse reward tasks more efficient. They propose a method called Workflow Guided Exploration (WGE) which is learnt from demonstrations but is environment agnostic. Episodes are generated by first turning demonstrations to a workflow lattice. This lattice encodes actions which are in some sense similar to those in the demonstration. By rolling out episodes which are randomly sampled from this set of similar actions for each encountered state, it is claimed that other methods like Behavor Cloning + RL (BC-then-RL) can be outperformed in terms of number of sample complexity since high reward episodes can be sampled with much higher probability.

A novel NN architecture (DOMNet) is also presented which can embed structured documents like HTML webpages.

Comments:

- The paper is well-written and relevant literature is cited and discussed.
- My main concern is that while imitation learning and inverse reinforcement learning are mentioned and discussed in related work section as classes of algorithms for incorporating prior information there is no baseline experiment using either of these methods. Note that the work of Ross and Bagnell, 2010, 2011 (cited in the paper) establish theoretically that Behavior Cloning does not work in such situations due to the non-iid data generation process in such sequential decision-making settings (the mistakes grow quadratically in the length of the horizon). Their proposed algorithm DAgger fixes this (the mistakes by the policy are linear in the horizon length) by using an iterative procedure where the learnt policy from the previous iteration is executed and expert demonstrations on the visited states are recorded, the new data thus generated is added to the previous data and a new policy retrained. Dagger and related methods like Aggrevate provide sample-efficient ways of exploring the environment near where the initial demonstrations were given. WGE is aiming to do the same: explore near demonstration states.
- The problem with putting in the replay buffer only episodes which yield high reward is that extrapolation will inevitably lead the learnt policy towards parts of the state space where there is actually low reward but since no support is present the policy makes such mistakes.
- Therefore would be good to have Dagger or a similar imitation learning algorithm be used as a baseline in the experiments.
- Similar concerns with IRL methods not being used as baselines.

Update: Review score updated after discussion with authors below.

---

> ### Author Response · Authors · 2017-12-19
> **Reply to AnonReviewer1**
>
> We would like to thank the reviewer for the feedback!
>
> The reviewer suggested further comparisons with inverse reinforcement learning (IRL) and DAgger-based methods (e.g., DAgger, AggreVaTe). In our paper revision, we will address the critical differences between our setting and the settings of these methods, which are summarized below:
>
> - In IRL, the system does not receive rewards from the environment and instead extracts a reward function from demonstrations. In our setting, the system already observes the true reward from the environment, so applying IRL would be redundant. Furthermore, IRL would struggle to learn a good reward function from such a small number of demonstrations (e.g., 3-10), which we have in our setting.
>
> - DAgger-based methods require access to an expert policy, which is iteratively queried to augment the training data. In our setting, the system gets a small number of demonstrations and can interact with the environment, but does not have access to an expert policy, so these methods cannot be directly applied. In addition, while DAgger-based methods do indeed provide an alternative way to explore around a neighborhood of the demonstrations, their goal is different from ours: DAgger addresses compounding errors, while our work addresses finding sparse reward.
>
> Finally, we want to clarify the concern that only high reward episodes are placed in the buffer. The neural policy updates both off-policy from the buffer and on-policy during roll-outs. If the neural policy begins to make mistakes, it will be penalized by receiving low reward during the on-policy rollouts, which will correct these mistakes.

---

> > ### Comment · AnonReviewer1 · 2017-12-31
> > **Clarification question.**
> >
> > " In our setting, the system gets a small number of demonstrations and can interact with the environment, but does not have access to an expert policy, so these methods cannot be directly applied" --Aren't the demonstrations coming from a human? Then isn't the human the expert in this case?

---

> > > ### Author Response · Authors · 2017-12-31
> > > **Reply to the clarification question**
> > >
> > > Our system initially receives a fixed set of human demonstrations and afterward receives no human supervision. In contrast, DAgger and related methods repeatedly query the expert policy (the human) at states reached by the learned policy during training.
> > >
> > > In other words, in DAgger and related methods, the expert policy must be available throughout the entire training procedure, whereas in our work, we only roll out a fixed number of episodes from the expert policy at initialization, with no further access.

---

> > > > ### Comment · AnonReviewer1 · 2018-01-01
> > > > **Clarification contd..**
> > > >
> > > > But there is a 'schedule' of invoking the expert in the training process of DAgger and related methods where the expert invocation decays over iterations and in practice over all domains not more than five iterations are usually needed.
> > > >
> > > > Is there reason to believe that the total number of expert invocations in DAgger will be less than WGE for similar performance? This is exactly where an empirical comparison will be really useful and   seal the usefulness of WGE.

---

> > > > > ### Author Response · Authors · 2018-01-05
> > > > > **Reply to Clarification question**
> > > > >
> > > > > We have implemented DAgger with oracle expert labels on a total of 24 episodes (more than twice the number of demonstrations for WGE). DAgger successfully solves the click-checkboxes task, which is expected since BC+RL also solves the task. However, on the harder click-checkboxes-large task where WGE got 84% success rate, DAgger gets 0% success rate even when we run RL on top afterward. Increasing the limit on the number of expert labels to 100 episodes (10 times the demonstrations for WGE) increases the success rate of DAgger to 50%, which is still lower than WGE.
> > > > >
> > > > > In our experiments, behavioral cloning (BC) suffers from two problems: compounding errors and data sparsity (from learning from so few demonstrations). DAgger addresses the compounding errors problem, but does not address the data sparsity problem. We empirically observe that the neural policy typically requires over 1000 reward-earning episodes to successfully learn the harder tasks. This suggests that BC is failing mainly due to data sparsity (because we have few demonstrations available in our setting). This also explains why DAgger fails to learn the harder tasks, even with 100 demonstrations. In contrast, WGE succeeds in most harder tasks using only 10 demonstrations.

---

### Author Response · Authors · 2018-01-05
**Updated submission**

We have updated our submission to reflect the comments and suggestions by the reviewers. In particular, in the Discussion section, we have expanded the comparison to related work on learning from demonstrations. Additionally, we have highlighted the important results of our experiments and clarified a few confusing terms.

---

### Decision · Program_Chairs · 2018-01-29
**ICLR 2018 Conference Acceptance Decision**

**Decision:**

Accept (Poster)

**Comment:**


PROS:
1. well-written and clear
2. added extra comparison to dagger which shows success
3. SOTA results on open ai benchmark problem and comparison to relevant related work (Shi 2017)
4. practical applications
5. created new dataset to test harder aspects of the problem

CONS:
1. the algorithmic novelty is somewhat limited
2. some indication of scalability to real-world tasks is provided but it is limited